# Information-Theoretic Evaluation of Free-Text Rationales with Conditional $\mathcal{V}$-Information

**Hanjie Chen**$^{\heartsuit *}$, **Faeze Brahman**$^{\spadesuit,\diamond}$, **Xiang Ren**$^{\clubsuit}$,
**Yangfeng Ji**$^{\heartsuit}$, **Yejin Choi**$^{\spadesuit,\diamond}$, **Swabha Swayamdipta**$^{\clubsuit}$

$^{\heartsuit}$Department of Computer Science, University of Virginia
$^{\spadesuit}$Allen Institute for Artificial Intelligence; $^{\clubsuit}$University of Southern California
$^{\diamond}$Paul G. Allen School of Computer Science, University of Washington
{hc9mx, yangfeng}@virginia.edu; faezeb@allenai.org;
{swabhas, xiangren}@usc.edu; yejin@cs.washington.edu

## Abstract

Free-text rationales are a promising step towards explainable AI, yet their evaluation remains an open research problem. While existing metrics have mostly focused on measuring the direct association between the rationale and a given label, we argue that an ideal metric should also be able to focus on the new information uniquely provided in the rationale that is otherwise not provided in the input or the label. We investigate this research problem from an information-theoretic perspective using the conditional $\mathcal{V}$-information [Hewitt et al., 2021]. More concretely, we propose a metric called REV (Rationale Evaluation with conditional $\mathcal{V}$-information), that can quantify the new information in a rationale supporting a given label *beyond* the information already available in the input or the label. Experiments on reasoning tasks across four benchmarks demonstrate the effectiveness of REV in evaluating different types of rationale-label pairs, compared to existing metrics. Furthermore, REV is consistent with human judgments on rationale evaluations. Overall, when used alongside traditional performance metrics, REV provides deeper insights into a models' reasoning and prediction processes.

## 1 Introduction

Model explanations have been indispensable for trust and interpretability in AI [Lipton, 2018, Doshi-Velez and Kim, 2017, Kim et al., 2016, Alvarez Melis and Jaakkola, 2018]. Free-text rationales, which explain a model prediction in natural language, have been especially appealing due to their flexibility in eliciting the reasoning process behind the model's decision making [Camburu et al., 2018, Narang et al., 2020, Rajani et al., 2019, Kumar and Talukdar, 2020b, Brahman et al., 2021], making them closer to human explanations. However, current automatic evaluation of free-text rationales remains narrowly focused. Existing metrics primarily measure the extent to which a rationale can help a (proxy) model predict the label it explains (i.e., accuracy based) [Hase et al., 2020, Wiegreffe et al., 2021]. Yet, these metrics offer little understanding of the *new information* contained in the rationale, as added to the original input, that could *explain why the label is selected*—the very purpose a rationale is designed to serve. For instance, the two rationales $r_1^*$ and $\hat{r}_{1,a}$ in Fig. 1 would be considered equally valuable under existing metrics, even though they supply different amount of novel and relevant information.

In this paper, we overcome this shortcoming by introducing an automatic evaluation for free-text rationales along two dimensions: (1) whether the rationale supports (i.e., is predictive of) the in-

---

$^{*}$Work done during internship at AI2.

36th Conference on Neural Information Processing Systems (NeurIPS 2022).

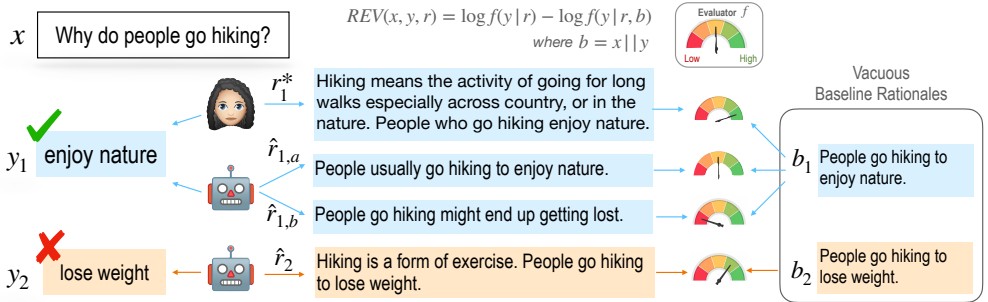

Figure 1: Our evaluation framework for different free-text rationales ($r$). $r_1^*$ is a human-written rationale, $\hat{r}_{1,a}$ and $\hat{r}_{1,b}$ are two generated rationales for the true label $y_1$. Our metric, REV, based on CVI [Hewitt et al., 2021] is able to distinguish all three rationales by measuring how much new and relevant information each adds over a vacuous rationale, $b$; performance-based evaluations can only distinguish between $\hat{r}_{1,a}$ and $\hat{r}_{1,b}$. For an (arguably) incorrect label, $y_2$, REV still gives a positive score highlighting that $\hat{r}_2$ is able to provide new information for why it supports $y_2$. Prediction accuracy can be augmented with REV to provide a fuller interpretability of model decisions.

tended label, and (2) how much *new information* does it provide to justify the label, **beyond** what is contained in the input. For example, rationale $\hat{r}_{1,b}$ in Fig. 1 violates (1) because it is not predictive of the label, "enjoy nature". Rationale $\hat{r}_{1,a}$ does support the label but contains no new information that justifies it, *beyond* what is stated in the input $x$; thus, it violates (2). Rationale $r_1^*$ is satisfied along both dimensions: it supports the label and does so by providing new and relevant information, beyond what is in the input. Our proposed evaluation is designed to penalize both $\hat{r}_{1,a}$ and $\hat{r}_{1,b}$, while rewarding rationales like $r_1^*$.

We introduce a REV[2], which adapts an information-theoretic framework from Xu et al. [2019] for evaluating free-text rationales along the two dimensions mentioned above. Specifically, REV is based on conditional $\mathcal{V}$-information [Hewitt et al., 2021], which quantifies the degree of information contained in a representation, *beyond* another (baseline) representation, accessible to a model family, $\mathcal{V}$. As our baseline representation, we consider any vacuous rationale which simply combines an input with a given label, without providing any new information relevant to answering why the label was chosen. REV adapts conditional $\mathcal{V}$-information to evaluate rationales, where the representation is obtained via an evaluator model trained to produce a label given the rationale. Other metrics do not take into consideration vacuous rationales, and are hence unable to measure new, label-relevant information in rationales, beyond a vacuous baseline.

Our experiments present evaluations with REV for rationales under two reasoning tasks, common-sense question-answering (CQA; Talmor et al., 2018) and natural language inference (NLI; Bowman et al., 2015), across four benchmarks. Several quantitative evaluations demonstrate the capabilities of REV in providing evaluations along new dimensions for free-text rationales, while being more consistent with human judgements compared to existing metrics.

## 2 REV: Information-Theoretic Evaluation of Rationales

We introduce a new metric, REV, Rationale Evaluation with conditional $\mathcal{V}$-information, for evaluation of free-text rationales on the proposed dimensions (§2.2), based on the information-theoretic framework of conditional $\mathcal{V}$-information (§2.1).

We consider the setting where we have input $X \in \mathcal{X}$, label $Y \in \mathcal{Y}$, and free-text rationale $R \in \mathcal{R}$ generated for label $Y$. A common strategy to evaluate rationales $R$ is through an evaluator $f \in \mathcal{V}$ based on how much $R$ helps $f$ predict $Y$ given $X$. The evaluator $f : Z \rightarrow Y$ maps a variable $Z$ to a label distribution. The definition of $Z$ depends on the evaluation framework; e.g., $Z$ can be a concatenation of $X$ and $R$. The evaluator $f$ is trained on a set of input, output and rationale triples $\mathcal{D}_{\text{train}} = \{(x_j, y_j, r_j)\}$, and applied to $\mathcal{D}_{\text{test}} = \{(x_i, y_i, r_i)\}$ for evaluation. The utility of $R$ is formulated as the difference between the performance of the evaluator on predicting $Y$ with $R$, and

---

[2]For Rationale Evaluation with conditional $\mathcal{V}$-information.

without it, i.e.
$$\text{Perf}[f(Y|X, R)] - \text{Perf}[f(Y|X)]. \tag{1}$$
A larger performance gap indicates a better rationale. Existing metrics [Hase et al., 2020, Wiegreffe et al., 2021] compute the performance gap based on prediction accuracies, measuring how much $R$ can help the evaluator correctly predict $Y$ given $X$.

However, accuracy-based evaluation can only indicate whether or not a rationale is predictive of a label, but cannot quantify how much *new information the rationale provides to justify the label*. Fig. 1 illustrates this issue via an example. Accuracy-based evaluation can distinguish between $\hat{r}_{1,a}$ and $\hat{r}_{1,b}$ since $\hat{r}_{1,a}$ supports $y_1$ and $\hat{r}_{1,b}$ does not. However, it is unable to distinguish between $r_1^*$ and $\hat{r}_{1,a}$ (since both are predictive of $y_1$), despite the fact that $\hat{r}_{1,a}$ does not provide any unique and relevant information to answer why the label should be $y_1$. In practice, vacuous rationales such as $\hat{r}_{1,a}$ are commonly seen in model generations [Sun et al., 2022, Wiegreffe and Marasović, 2021]. This calls for an evaluation metric which is able to identify and penalize such vacuous rationales.

## 2.1 An Information-Theoretic Perspective on Rationale Evaluation

The key quantity of interest for our evaluation of rationales $R$ is the amount of new information expressed in $R$ (e.g., background knowledge, reasoning process) that can justify a label $Y$. The mutual information between $R$ and $Y$, $I(Y; R)$ can be helpful for evaluating this quantity. However, we are not interested in the information that is already captured in the input $X$. A **vacuous** rationale, such as $\hat{r}_{1,a}$ in Fig. 1, which simply combines the input $X$ and the label, $Y$, captures all the information in $X$ and $Y$ without specifying any new information to help understand why $Y$ has been chosen for $X$; let us denote such rationales as $B \in \mathcal{B}$. Thus, we argue that a good evaluation metric must be able to measure the amount of relevant, new information contained in a rationale *beyond* what is contained in any vacuous rationale, $B$, that leads to the prediction of $Y$. Then the new information in $R$ beyond what is available in $B$ can be grounded with conditional mutual information [Shannon, 1948] as follows,
$$I(Y; R \mid B) = I(Y; R, B) - I(Y; B), \tag{2}$$
where the difference of two information quantities demonstrates the performance gap in Equation 1. Directly computing mutual information, however, is challenging because true distributions of random variables are usually unknown, and we do not have unbounded computation. A recently introduced information-theoretic framework called $\mathcal{V}$-information circumvents this by restricting the computation to certain predictive model families, $\mathcal{V}$ [Xu et al., 2019]. Our approach to evaluate rationales extends this framework, following [Hewitt et al., 2021], as described below.

**Conditional $\mathcal{V}$-information**   Given a model family $\mathcal{V}$ that maps two random variables $R$ and $Y$, $\mathcal{V}$-information defines the usable information that can be extracted from $R$ by models in $\mathcal{V}$ to predict $Y$, i.e. $I_{\mathcal{V}}(R \rightarrow Y)$. If $\mathcal{V}$ generalizes to the set of all possible functions, then $\mathcal{V}$-information is mutual information [Shannon, 1948]. In practice, it is feasible to estimate the usable information from $R$ about $Y$ by selecting any neural model without frozen parameters as $\mathcal{V}$.[3]

Following conditional mutual information in information theory [Cover and Thomas, 2006], $\mathcal{V}$-information has been extended to conditional $\mathcal{V}$-information (CVI; Hewitt et al., 2021). CVI quantifies the $\mathcal{V}$-usable information in $R$ about $Y$ conditioned on a variable $B$, i.e.
$$I_{\mathcal{V}}(R \rightarrow Y \mid B) = H_{\mathcal{V}}(Y \mid B) - H_{\mathcal{V}}(Y \mid R, B). \tag{3}$$
Here $B$ is any vacuous rationale that leads to the prediction of $Y$. In this work, we consider $B$ simply as the concatenation of $X$ and $Y$. We leave analyzing how different baseline construction impacts our metric to future work. $H_{\mathcal{V}}(\cdot \mid \cdot)$ is the conditional $\mathcal{V}$-entropy [Xu et al., 2019, Hewitt et al., 2021, Ethayarajh et al., 2022], defined as
$$H_{\mathcal{V}}(Y \mid B) = \inf_{f \in \mathcal{V}} \mathbb{E}[-\log f[b](y)]; \quad H_{\mathcal{V}}(Y \mid R, B) = \inf_{f \in \mathcal{V}} \mathbb{E}[-\log f[r, b](y)], \tag{4}$$
where $f[b]$ and $f[r, b]$ produce a probability distribution over the labels given $b$ and $[r, b]$ as inputs respectively.[4] Further, we consider pointwise CVI for evaluating individual samples, $(r, y, b)$ as
$$PI_{\mathcal{V}}(r \rightarrow y \mid b) = \log f[r, b](y) - \log f[b](y). \tag{5}$$

---

[3]Please see [Xu et al., 2019] for a detailed discussion of properties such as optional ignorance that a predictive family $\mathcal{V}$ must follow.

[4]Please see Appendix A for further details on CVI.

## 2.2 Computing REV for Rationale Evaluation

Building on the framework of CVI, we propose a new metric REV, for Rationale Evaluation with conditional $\mathcal{V}$-information. We compute REV over a given test set, $\mathcal{D}_{\text{test}} = \{(x_i, y_i, r_i)\}$, by estimating CVI over the set with an evaluator $f \in \mathcal{V}$. For a test example $(x, y, r)$, the REV score denoted as $\text{REV}(x, y, r)$ is computed based on Equation 5, where $b$ is constructed by combining $x$ and $y$.

$$\text{REV}(x, y, r) = PI_{\mathcal{V}}(r \to y \mid b) \tag{6}$$

The REV score for the test corpus $\mathcal{D}_{\text{test}}$, is given by the average pointwise REV score:

$$\text{REV} = \frac{1}{|\mathcal{D}_{\text{test}}|} \sum_i \text{REV}(x_i, y_i, r_i). \tag{7}$$

The higher the REV score, the more additional (*new* and *relevant*) information the rationale $r$ contains to explain the label beyond the baseline rationale $b$. $\text{REV}(x, y, r)$ can take positive, negative, or zero values. When $\text{REV}(x, y, r)) > 0$, the rationale supplies additional information for supporting the label (e.g., $r_1^*$ in Fig. 1); when $\text{REV}(x_i, y_i, r_i) = 0$, the rationale provides no additional information beyond the baseline (e.g., $r_{1,a}$ in Fig. 1); and when $\text{REV}(x_i, y_i, r_i) < 0$, the rationale contains additional information which does *not* support the label (e.g., $r_{1,b}$ in Fig. 1). REV can assign a positive score to a rationale for an incorrect prediction as long as the rationale supports it and provides additional information beyond a vacuous baseline rationale. Thus, REV cannot be seen as a replacement for prediction accuracy, but rather as an orthogonal metric to interpret the usefulness of a generated rationale for the model decision.

**Constructing a Baseline with Vacuous Rationales**  Given an input $x$ and a label $y$, we construct a baseline rationale $b$ by converting $x$ and $y$ into a declarative sentence. For the CQA task, we adopt a pre-trained T5-3B model fine-tuned on a set of (*question*, *answer*, *declarative sentence*) tuples [Chen et al., 2021] annotated by Demszky et al. [2018]. For the NLI task, we use a template to convert (*premise*, *hypothesis*, *label*) tuple into a baseline rationale: "*premise* `implies` / `contradicts` / `is not related to` *hypothesis*". Table 2 shows some examples of constructed vacuous rationales.

**Training the evaluator, $f$**  We select a generative model $f \in \mathcal{V}$ as the evaluator to learn the mapping $f : [r, b] \to y$, where the input $[r, b]$ is the concatenation of $r$ and $b$.[5] In particular, we use pre-trained language models [e.g., T5; Raffel et al., 2020] and fine-tune them on the training set $\mathcal{D}_{train} = \{(x, y^*, r^*)\}$, where $\{y^*\}$ and $\{r^*\}$ are gold labels and human-annotated rationales, respectively. We construct baseline rationales $\{b^*\}$ based on $\{(x, y^*)\}$. The objective is to maximize the log-likelihood of $y^*$ given $r^*$ and $b^*$.

After training, the evaluator can be applied to evaluate a given rationale-label pair $(y, r)$ w.r.t. an input $x$. The rationale-label pair $(y, r)$ can be model-generated and the label may not be ground-truth (e.g., $y_2$ in Fig. 1), while REV is still able to provide an assessment on the rationale along the two dimensions (§1), e.g., $\text{REV}(x, y_2, \hat{r}_2) = 0.6$ in Fig. 1.

## 3 Experimental Setup

We outline our experimental setup by describing the reasoning tasks and datasets (§3.1), followed by the task and evaluation models (§3.2), and the baseline metrics for comparison (§3.3). Additional details on the setup are provided in Appendix C.

### 3.1 Datasets

We explore CommonsenseQA (CQA) across 3 datasets, all containing human-annotated free-text rationales. For CQA task, we use ECQA [Aggarwal et al., 2021], CoS-E (v1.11; Rajani et al., 2019) and QuaRTz [Tafjord et al., 2019]. ECQA contains higher quality human-written rationales compared to CoS-E [Aggarwal et al., 2021] More details of the datasets are in Appendix C.2.

---

[5]We do not train two models as Hewitt et al. [2021] did, one taking as input $[r, b]$ and the other taking as input $b$ padded with dummy tokens. In our pilot experiments, the model trained solely with $b$ can be overconfident on its predictions as $b$ simply leaks the label information.

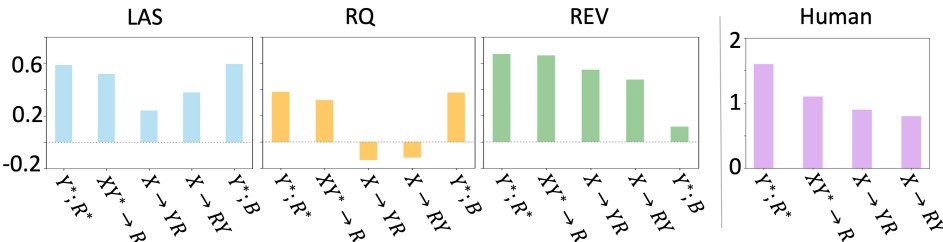

Figure 2: Left: automatic evaluation results of LAS, RQ and REV for rationale-label pairs on the ECQA test set. Right: human evaluation for rationale-label pairs on 230 randomly selected examples from the ECQA test set.

## 3.2 Task and Evaluation Models

**Task models** We choose T5 Large [Raffel et al., 2020] as the task model (finetuned on ground truth labels and rationales) to produce generated rationale-label pairs under three settings:

- $XY^*{\rightarrow}R$: Given an input text and the gold label, generate a rationale.
- $X{\rightarrow}YR$: Given an input text, generate a label followed by a rationale. Since T5 decodes tokens sequentially, each R is generated conditioned on the predicted Y.
- $X{\rightarrow}RY$: Given an input text, generate a rationale followed by a label. Here, we compute a likelihood for each candidate Y conditioned on R, and then select the most probable candidate. This operation can improve the model prediction accuracy, while weakening the consistency and relevance between the generated rationales and predicted labels.

After training, we collect three types of rationale-label pairs by applying the three task models on the test set of each dataset. In addition to these three settings, we also evaluate ground-truth labels paired with crowd-sourced rationales $(Y^*;R^*)$.

**Evaluators** Our evaluator, $f$ (see Equation 6 in §2) is also based on T5 Large trained on gold rationale-label pairs of the respective dataset. We refer readers to the Appendix D.1 for results using T5 Base, BART Large [Lewis et al., 2020] and GPT-2 Large [Radford et al., 2019] as the evaluator.

## 3.3 Other Metrics for Rationale Evaluation

We compare with two existing automatic metrics for free-text rationale evaluation: LAS [Hase et al., 2020] and RQ [Wiegreffe et al., 2021]. Analogous to our evaluator, $f$, both approaches use proxy models; we use the same architecture (T5 Large) across metrics in our reported results. LAS computes the difference between the prediction accuracy of a model proxy on the predicted label when the rationale is included into the input vs. when it is not, $\mathbb{1}[\hat{y} \mid x, \hat{r}] - \mathbb{1}[\hat{y} \mid x]$, averaged over examples grouped based on whether they leak labels or not. Differently, RQ uses gold labels. The quality of a rationale $\hat{r}$ is measured as $\mathbb{1}[y^* \mid x, \hat{r}] - \mathbb{1}[y^* \mid x]$, where $y^*$ is the gold label. Similarly, RQ is the average score over all test examples.

## 4 Experiments

We compare REV with existing metrics (§4.1) and human judgments (§4.2) on the ECQA dataset, as well as show REV on other CQA and NLI benchmarks.

## 4.1 Comparison Between Evaluation Metrics

We compare REV to LAS and RQ, in evaluating different rationale-label pairs on the ECQA dataset. In addition to $Y^*;R^*$, $XY^*{\rightarrow}R$, $X{\rightarrow}YR$ and $X{\rightarrow}RY$, we also explore the evaluation on vacuous baseline rationales $(Y^*;B)$, which simply combine inputs and labels with no additional information. Note that the scores obtained from different metrics are not directly comparable due to different comparison scales and criteria (e.g., log-probability vs. accuracy). We mainly focus on the ranking

over different types of rationale-label pairs. The results averaged over 4 random seeds are shown in the left part of Fig. 2. Qualitative results are provided in Table 5 in Appendix D.2.

All three metrics agree that the crowdsourced rationales ($Y^*;R^*$) in the ECQA have the highest quality. While by definition, REV for vacuous rationales is low, both LAS and RQ scores for these rationales are quite high, showing that these metrics are incapable of measuring the amount of additional information in rationales. Intuitively, we expect weaker rationale-label consistency in $X{\to}RY$ setting compared to $X{\to}YR$, as the labels are forcefully selected among the candidates as opposed to being freely generated by the task model (§3.2). While REV is able to capture this intuition and rank $X{\to}YR$ higher than $X{\to}RY$, LAS and RQ have a different ranking.

Next, we apply REV to evaluate crowd-sourced and model generated rationale-label pairs ($Y^*;R^*$, $XY^*{\to}R$, $X{\to}YR$, $X{\to}RY$) across different datasets. For each dataset, the evaluator is trained on the training set with gold labels and crowdsourced rationales. The results are shown in Table 1. We observe that the gold rationales in the ECQA dataset achieve higher REV score than those in CoS-E. This observation is in line with the known quality issues of crowdsourced rationales in CoS-E [Aggarwal

| Datasets | Rationale-label pairs | | | |
|---|---|---|---|---|
| | $Y^*;R^*$ | $XY^*{\to}R$ | $X{\to}YR$ | $X{\to}RY$ |
| ECQA | 0.6684 | 0.6401 | 0.5285 | 0.4586 |
| CoS-E | 0.3476 | 0.4576 | 0.3328 | 0.1518 |
| QuaRTz | 0.1851 | 0.1896 | 0.1624 | 0.1572 |
| e-SNLI | 1.1e-6 | 1.1e-6 | 1.08e-6 | 1.09e-6 |

Table 1: REV scores of different types of rationale-label pairs on the four datasets.

et al., 2021, Sun et al., 2022]. Moreover, training the evaluator with CoS-E results in lower REV for all models, compared to training with ECQA. Interestingly, model-generated rationales ($XY^*{\to}R$) have higher REV scores than crowdsourced rationales for CoS-E, and similar REV scores for ECQA, QuaRTz and e-SNLI. For QuaRTz, the model generated rationales seem to not contain much new and relevant information over a vacuous baseline. In the case of e-SNLI, the problem is even severer as most of the crowdsourced or generated rationales do not provide reasoning but rather follow a label-specific template e.g., *A implies (that) B* [Kumar and Talukdar, 2020a, Brahman et al., 2021].

## 4.2   Human Evaluation

To understand how REV correlates with human judgments of rationales, we conduct a crowdsourcing experiment via Amazon Mechanical Turk. We randomly sample 230 examples from the ECQA test set and ask workers to evaluate the four types of rationale-label pairs ($Y^*;R^*$, $XY^*{\to}R$, $X{\to}YR$, $X{\to}RY$) for each example. We present workers with a question (input text), an answer (label) and an explanation (rationale), and ask them whether the explanation justifies the answer (*yes/no*). If they answer *yes*, we further ask them to evaluate the amount of additional information supplied by the explanation that explains *why* the answer might have been chosen for the question. The workers choose from *none / little / some / enough*, corresponding to a 4-point Likert-scale. We collect 3 annotations per instance and use majority vote to decide whether the rationale can justify the label. If *yes*, we take the average over the 3 human-annotated scores as the amount of information. Otherwise, we give a score of -1. More details of human evaluation are in Appendix D.3.

The results are shown in the right part of Fig. 2, where the ranking of the four types of rationale-label pairs is $Y^*;R^* > XY^*{\to}R > X{\to}YR > X{\to}RY$. While LAS and RQ rank $X{\to}RY$ better than $X{\to}YR$ (see the left part of Fig. 2), the ranking from REV is more consistent with human judgments, suggesting its effectiveness in evaluating rationale quality.

## 5   Related Work

Self-rationalized models serve interpretability by providing rationales for their predictions. The rationales broadly fall into two categories: extractive rationales and free-text rationales. Evaluations on extractive rationales have been well studied, generally from two perspectives — faithfulness and plausibility [DeYoung et al., 2020, Pruthi et al., 2022, Chan et al., 2022b]. Faithfulness measures to which extent rationales reflect the true reasoning process of models, while plausibility evaluates how convincing rationales are to humans [Jacovi and Goldberg, 2020]. However, the faithfulness evaluation metrics (e.g., sufficiency, comprehensiveness) proposed for extractive rationales are not applicable to free-text rationales because they are not a part of inputs. Existing automatic metrics for free-text rationales focus on rationale-label association, and measure the utility of a rationale based

on how much it helps a model proxy predict the given label [Hase et al., 2020] or the gold label [Wiegreffe et al., 2021] given the input. Chan et al. [2022a] further propose a framework to evaluate the automatic metrics. However, none of them consider measuring the amount of additional information in free-text rationales. Sun et al. [2022] conduct a human study on the additional knowledge provided by free-text rationales. This work is the first that proposes an automatic metric to quantify the additional information in free-text rationales.

## 6    Conclusion

In this paper, we propose an information-theoretic metric, REV, to evaluate free-text rationale. REV measures if a rationale contains new information that is relevant for the label of interest, beyond the information contained in the input. We show the advantage of REV in evaluating different types of rationale-label pairs compared to existing metrics. We demonstrate that the evaluation of free-text rationales with REV is consistent with human judgments. Future work might explore evaluation that penalizes rationales which support incorrect predictions, thus bridging together predictive performance with interpretability metrics.

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

# A  Properties of Conditional $\mathcal{V}$-information

As proved by Hewitt et al. [2021], CVI has several useful properties:

1. *Non-Negativity*: $I_{\mathcal{V}}(R \to Y \mid B) \geq 0$.
2. *Independence*: If $Y$ and $B$ are jointly independent of $R$, then $I_{\mathcal{V}}(R \to Y \mid B) = 0$.
3. *Monotonicity*: If $\mathcal{U} \subseteq \mathcal{V}$, then $H_{\mathcal{V}}(Y \mid B) \leq H_{\mathcal{U}}(Y \mid B)$.

An implication from *Monotonicity* is complex models (e.g., pre-trained language models) can do better than simpler ones (e.g., linear models) in estimating $\mathcal{V}$-usable information. Since CVI measures the additional $\mathcal{V}$-usable information in $R$ about $Y$ beyond what's already extracted from $B$ by models in $\mathcal{V}$, it grounds the goal of the proposed metric REV.

# B  Computing REV for Rationale Evaluation

Algorithm 1 shows the process of computing both pointwise and aggregate REV scores.

# C  Supplement of Experimental Setup

## C.1  Constructing a Baseline with Vacuous Rationales

| Task | Input | Label | Baseline Rationale |
|------|-------|-------|--------------------|
| CQA | Where can personal mushrooms be kept fresh? | refrigerator | Personal mushrooms can be kept fresh in the refrigerator. |
| NLI | Premise: A dog running in the surf. Hypothesise: A dog is at the beach. | entailment | A dog running in the surf implies a dog is at the beach. |

Table 2: Examples of constructed baseline rationales for CQA and NLI tasks.

## C.2  Datasets

For CQA task, we use ECQA [Aggarwal et al., 2021], CoS-E (v1.11) [6] [Rajani et al., 2019] and QuaRTz [Tafjord et al., 2019]. Both ECQA and CoS-E originate from the CommonsenseQA dataset [Talmor et al., 2018], where each commonsense question is paired with 5 candidate choices and the task is to select an answer from the candidates. ECQA contains higher quality free-text rationales compared to CoS-E, in terms of comprehensiveness, coherence, non-redundancy, etc. [Aggarwal et al., 2021, Sun et al., 2022]. QuaRTz is an open-domain reasoning task about textual qualitative relationships. Each instance contains a situated qualitative question, two answer options and a knowledge statement. The task is to select an answer from the two options to the question based on the textual qualitative knowledge. We use the knowledge statement as a free-text rationale since it explains why the answer is to the question. For NLI task, we use e-SNLI [Camburu et al., 2018] which is an extension of SNLI

---

**Algorithm 1** Computing REV Scores

1: **Input**: evaluator $f$, test set $\mathcal{D}_{\text{test}} = \{(x_i, y_i, r_i)\}$
2: Initialize an empty set $\mathcal{S} = \varnothing$
3: $\text{REV} \leftarrow 0$
4: **for** $(x_i, y_i, r_i) \in \mathcal{D}_{\text{test}}$ **do**
5:     Construct the baseline rationale $b_i$
6:     $\text{REV}(x_i, y_i, r_i)$
       $= \log f[r_i, b_i](y_i) - \log f[b_i](y_i)$
7:     $\mathcal{S} \leftarrow \mathcal{S} \cup \text{REV}(x_i, y_i, r_i)$
8:     $\text{REV} \leftarrow \text{REV} + \text{REV}(x_i, y_i, r_i)$
9: **end for**
10: $\text{REV} \leftarrow \text{REV} / |\mathcal{D}_{\text{test}}|$
11: **Output**: $\mathcal{S}$, $\text{REV}$

---

[Bowman et al., 2015] with augmented free-text human-written rationales. The task is to predict the entailment relationship between a premise and a hypothesis. Table 3 shows the summary statistics of the four datasets. [7]

---

[6] We use the version v1.11 where each question is paired with 5 answer choices, for comparison with ECQA.

[7] Since CoS-E does not provide rationales for instances in the test set, we use the original development set as the test set and hold out 10% of training data as the new development set. We follow Hase et al. [2020] and randomly sample 10% of training data to form the training set for finetuning our models.

| Datasets | #train | #dev | #test |
|----------|--------|------|-------|
| ECQA     | 7598   | 1090 | 2194  |
| CoS-E    | 8766   | 975  | 1221  |
| QuaRTz   | 2696   | 384  | 784   |
| e-SNLI   | 54933  | 9842 | 9824  |

Table 3: Summary statistics of the datasets, where *#* counts the number of examples in the *train/dev/test* sets.

## C.3  Models

We use Huggingface Transformers [Wolf et al., 2020] to access all task models and evaluators. We train each model for up to 30 epochs with a learning rate $5e-6$ and a batch size $8$. All experiments were performed on a single NVIDIA RTX 8000 GPU. Table 4 shows input-output formattings of different task models for different tasks.

| Type | Input | Output |
|------|-------|--------|
| $XY^*{\rightarrow}R$ | CQA: [question] question [choice] choice-1 ... [choice] choice-n [answer] gold label [rationale] 
 NLI: [premise] premise [hypothesis] hypothesis [answer] gold label [rationale] | rationale <eos> |
| $X{\rightarrow}YR$ | CQA: [question] question [choice] choice-1 ... [choice] choice-n [answer] 
 NLI: [premise] premise [hypothesis] hypothesis [answer] | label [rationale] rationale <eos> |
| $X{\rightarrow}RY$ | CQA: [question] question [choice] choice-1 ... [choice] choice-n [rationale] 
 NLI: [premise] premise [hypothesis] hypothesis [rationale] | rationale [answer] label <eos> |

Table 4: The input-output formatting of different task models.

# D  Supplement of Experiments

## D.1  Comparison Between Evaluator Architectures

We apply REV to evaluate different types of free-text rationales w.r.t. labels on the ECQA dataset. Figure 3 shows REV scores of the four types of rationale-label pairs evaluated by the four evaluators. The ranking of the four groups of rationale-label pairs is consistent across the four evaluators, i.e. $Y^*;R^* > XY^*{\rightarrow}R > X{\rightarrow}YR > X{\rightarrow}RY$. This ranking is also consistent with human evaluation in §4.2. Since ECQA contains high-quality crowdsourced rationales [Aggarwal et al., 2021], it is expected that the REV of gold rationale-label pairs ($Y^*;R^*$) is the highest. The REV of $XY^*{\rightarrow}R$ is close to that of $Y^*;R^*$, indicating the task model (T5 Large) can produce good quality rationales when it is prompted with ground-truth labels. All four evaluators agree that the generated rationales of $X{\rightarrow}YR$ contain more additional background information for explaining the predicted labels than those of $X{\rightarrow}RY$. This is consistent with our design of the $X{\rightarrow}RY$ in §3.3, where the generated rationales and labels have weakened relevance. For each type of rationale-label pairs, the four evaluators capture different amount of conditional $\mathcal{V}$-information, while T5 Large consistently outperforms other three models. In the reported experiments §4, we use T5 Large as the evaluator.

## D.2  Qualitative Analysis of Different Metrics

Table 5 shows the qualitative analysis of different metrics on the four types of rationale-label pairs ($Y^*;R^*$, $XY^*{\rightarrow}R$, $X{\rightarrow}YR$, $X{\rightarrow}RY$) on the ECQA dataset. REV provides more accurate evaluations on those examples than LAS and RQ.

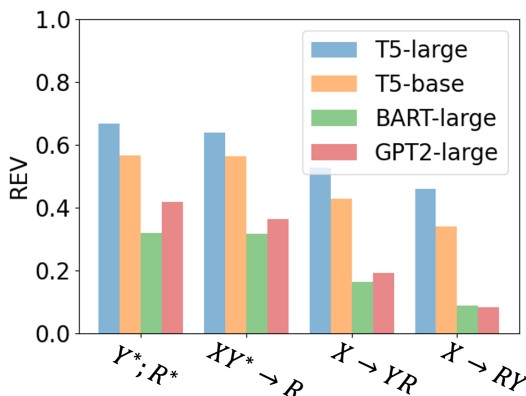

Figure 3: REV for evaluating rationale-label pairs on the ECQA dataset with different evaluator architectures.

### D.3 Human Evaluation Details

We randomly select 230 examples from the ECQA test set and conduct human evaluation on the four types of rationale-label pairs $(Y^*;R^*, XY^*{\rightarrow}R, X{\rightarrow}YR, X{\rightarrow}RY)$ w.r.t. each example through the Amazon Mechanical Turk (AMT). Each instance is assessed by 3 workers. We pay the workers $0.08 for assessing each instance.

Figure 4 shows the instructions we provide to workers. In Figure 5, we show three examples, illustrating when the explanation (rationale) does not justify the answer (label), when the explanation supports the answer while not supplying additional information, and when the explanation supports the answer and provides additional information. Figure 6 shows the interface of the actual hit for human evaluation.

For each instance, we provide a question (input), an answer (label), and an explanation (rationale), and ask the workers to answer the following two questions:

1. *Does the Explanation justify the given Answer?* (yes or no) The question is to ask workers to judge whether the rationale supports the label or not.

2. *If yes, how much additional information does the Explanation have to justify the Answer beyond just reiterating what is stated in Question and Answer?* (No additional info, Little additional info, Some additional info, Enough additional info) We only ask this question if the workers choose "yes" for the first question. We design this question to ask workers to evaluate the extent to which the rationale provides additional information for justifying the label beyond repeating it w.r.t. the input.

| Type | Question | Label | Rationale | Metric | | |
|---|---|---|---|---|---|---|
| | | | | REV | LAS | RQ |
| $Y^*;R^*$ | If you have a ticket and you are planning to eat hot dogs, where would you go? | baseball stadium | Hot dogs can be eaten at baseball stadium. When you go to a baseball stadium, you have a ticket and you may plan to eat hot dogs. | 0.13 | 0 | 0 |
| | How does a person go to space? | space shuttle | People go to space by a vehicle specially designed to travel to space. That vehicle is called a space shuttle. | 0.01 | 0 | 0 |
| | What is a dangerous outdoor activity for children? | sun themselves | Sunning themselves is a dangerous activity Children should not sun themselves | 0.02 | 1 | 1 |
| $XY^*{\rightarrow}R$ | Where are old pictures kept? | attic | Attic is a place where old pictures are kept. | 0.01 | 1 | 0 |
| | What would you be if you comfort friend? | friendly | Comforting friend is a good thing. | -0.03 | 0 | 1 |
| | What do customers do to a waiter after the waiter serves customers? | pay to | Paying to a waiter is the action of paying. Waiters get paid to serve customers. | 1.59 | -1 | 0 |
| $X{\rightarrow}YR$ | Where is there likely to b more than one desk drawer? | desk | Desk drawer is a drawer used for storing office supplies. There is likely to be more than one desk drawer in office. | -0.34 | -1 | 1 |
| | What leads to someone's death when they are very depressed? | suicide | Suicide is the act of committing suicide. When someone is very depressed, suicide leads to their death. | 0.25 | 0 | 0 |
| | Where are you normally when you take a bath? | hotel room | Hotel room is a place where people stay. Bathing is normally done in hotel rooms. | 0.01 | 0 | -1 |
| $X{\rightarrow}RY$ | What is likely heard by those going to a party? | laughter | People go to a party to meet new people. People are likely to hear laughter at the party. | 0.49 | 1 | 0 |
| | What would you do if you have excitement and do not want to stay in your house? | go to gym | Go to gym is to go to a place where you can express information. If you have excitement and do not want to stay in your house, then you would go somewhere. | 0.21 | 1 | 0 |
| | If you're caught committing murder, an injection can lead to your own what? | die | An injection can lead to one's own death. If you're caught committing murder, you can be injected into your own body and die. | 0.29 | 0 | 0 |

Table 5: Pointwise evaluation of REV, LAS and RQ on different types of rationale-label pairs. Incorrect labels are colored red.

**Instructions (click to expand/collapse)**

Thanks for participating in this HIT qualifier! Please read the examples below, then complete the below HIT (1-2 questions).

**Main Instructions**: you will read a question about daily life. For each question, an answer and a statement explaining the answer has been given.

**Your task** is to read the instance and answer questions about the explanation quality (1-2 questions). Regarding quality, you need to assess two aspects: (1) whether the explanation **supports the label**; (2) whether the explanation contains **additional information** for describing the answer beyond simply combining the question and the answer.

To be specific:
- "**supports the label**" means the explanation is describing something related to the answer to the question (e.g., Example #2 and #3 below), rather than something else (e.g., Example #1 below).

- "**additional information**" means the the explanation provides additional evidence or background knowledge to support the answer (e.g., Example #3 below), rather than simply combines the question and the answer (e.g., Example #2 below). You only evaluate the **additional information** when you agree that the explanation **supports the label** first. Please select the **amount of additional information** (from "No additional info" to "Enough additional info") based on the extent to which the explanation helps you understand *why* the answer might have been chosen for the question.

**An instance contains 3 parts:**

| | |
|---|---|
| *Question* | A question such as "*A hurricane is similar to what other wind event??*" |
| *Answer* | A selected answer, such as "*tornadoes*" (may or may not be correct). |
| *Explanation* | A **statement** which explains the *Answer*. |

NOTES
- **Important!** The *Answer* could be wrong and the *Explanation* might contain logical/grammatical mistakes. Please **ignore** these when answering the questions.

- You should judge whether the *Explanation* is describing the *Answer* and contains additional information to help you understand *why* the *Answer* might have been chosen for the *Question*.

Figure 4: The instructions of human evaluation in the user interface on AMT.

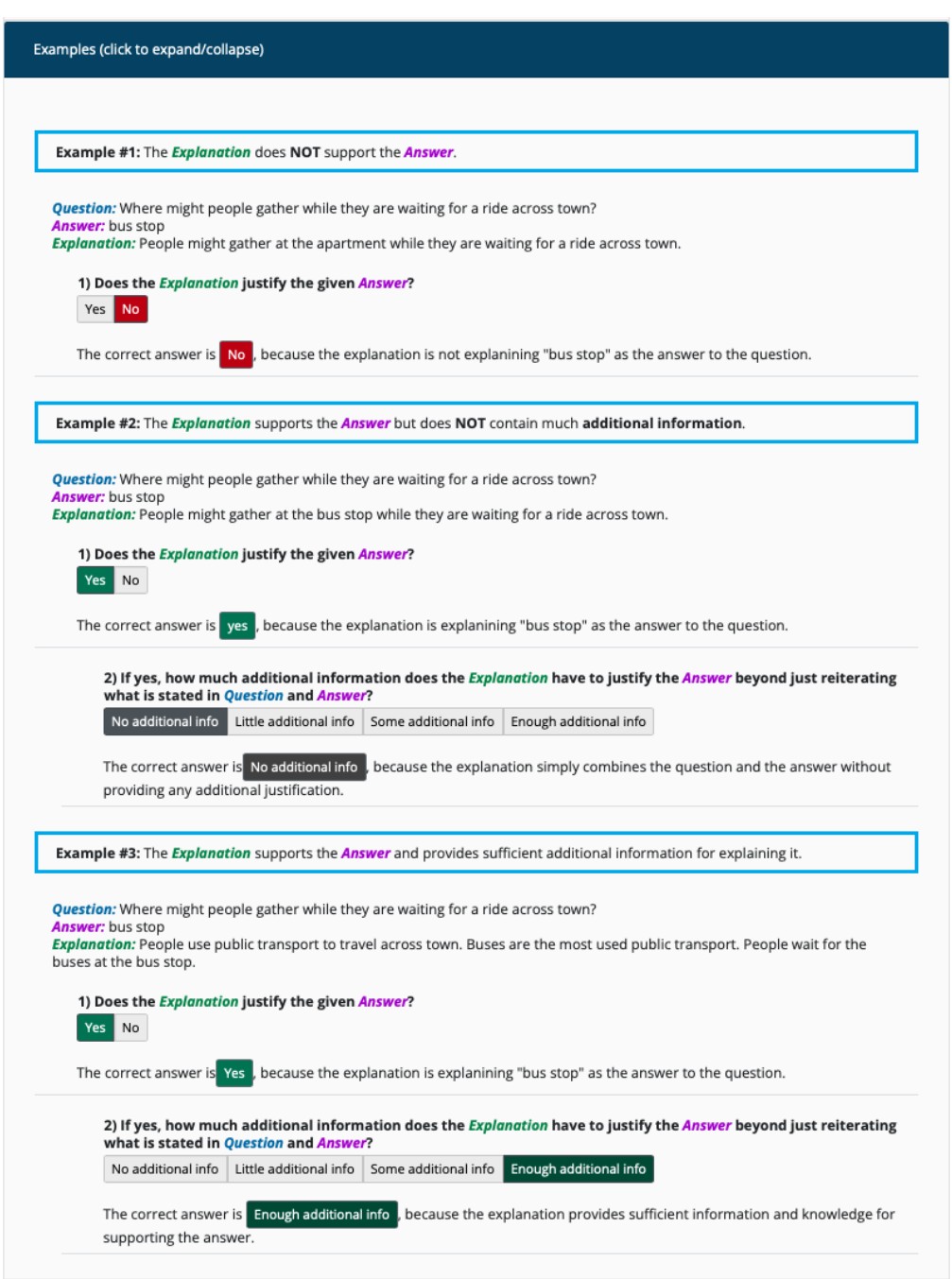

Figure 5: Exemplars provided to worker in the user interface on AMT.

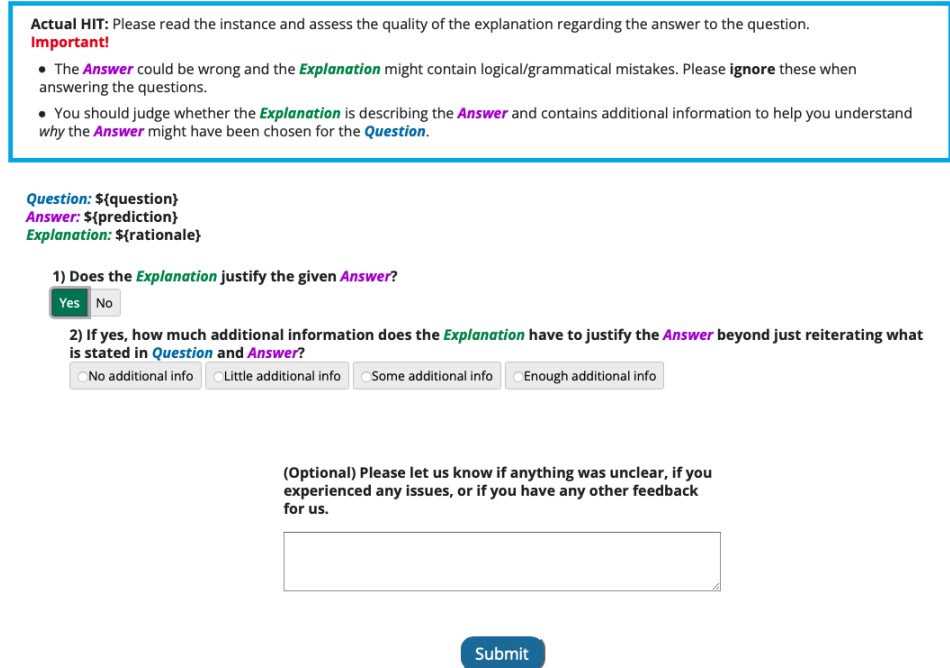

Figure 6: The actual hit of human evaluation in the user interface on AMT.

