# OpenReview forum: "Information-Theoretic Evaluation of Free-Text Rationales with Conditional $\mathcal{V}$-Information"
_NeurIPS.cc/2022/Workshop/TSRML — TSRML2022_

### Official Review · Reviewer_WR37 · 2022-10-10
**Well-written and interesting work addressing an important problem**

**Overall Rating:** 8

**Summary:**

Towards assessing the new relevant information provided by a free-text rationale (i.e., information not part of the input or label), the authors propose a metric called REV based on conditional V-information. The authors seek to show that REV is more effective than existing metrics for evaluating rationale-label pairs via experiments on four benchmarks and measurements of consistency with human judgments of rationale quality.

**Strengths:**

Quality:
- meanings of different REV values and orthogonality of REV to prediction accuracy gap are clearly explained
- convincing and well-explained experiments across different types of rationale-label pairs
- authors find higher REV scores for model-generated rationales (Table 2), insinuating crowdsourced rationales have low quality

Clarity:
- Figure 1 clearly illustrates need for REV and cases where REV improves over existing metrics
- theory is well-explained

Originality:
- existing rationale evaluation metrics focus on rationale-label association, but do not capture how much new and relevant information rationales provide over the input and label in explaining why the label was predicted
- interesting application of ${\cal V}$-information

Significance:
- free-text rationales are a valuable tool for understanding a model's decision-making process

**Weaknesses:**

Quality:
- how do $B = X$ and $B = Y$ separately affect conditional ${\cal V}$-entropy compared to $B = X || Y$?
- for experiments across different types of rationale-label pairs:
  - hard to compare results in Figure 2 because human and automatic evaluation metrics are on different scales; could the authors quantify the consistency of each metric with human judgment?
  - please include error bars in plots
  - why is there no human evaluation for $Y^*; B$?
  - would be better to separate and explain difference in results for ground-truth $Y^*$ vs. predicted $Y$ labels
- do the authors claim that human judgments of rationale quality are more aligned with rationale usefulness rather than rationales' predictive accuracy? how correlated are rationales' usefulness and predictive accuracy?
- how might this work be extended to measure the usefulness of rationales for the incorrectness (rather than correctness) of certain labels?

Clarity:
- Sections 1 and 2 have redundant content

Originality:
- work would benefit from further comparison to [1]; [1] uses ${\cal V}$-information to compute how much new and relevant information a training instance provides over other training instances

[1] https://proceedings.mlr.press/v162/ethayarajh22a.html

**Overall Recommendation:**

Well-written and interesting work that addresses the important problem of evaluating free-text rationales with a novel application of ${\cal V}$-usable information. Some suggestions for improving presentation of experimental results.

**Review Confidence:**

3: The reviewer is fairly confident that the evaluation is correct

---

### Official Review · Reviewer_zWUN · 2022-10-21

**Overall Rating:** 6

**Summary:**

In this paper, an information-theoretic metric based on recent V-information, called REV,  is proposed to evaluate free-text rationales. This metric can evaluate how much new information the rationales provide to justify the label, beyond what is included in the input. Experiments show that the evaluation of free-text rationales with REV is consistent with human judgments.

**Strengths:**

- Overall, the paper is well written and the motivation is clearly clarified.

- The proposed metric sounds well motivated by the real problem in free-text rationales.



**Weaknesses:**

One promising future work would be to explicitly incorporate desiderata into model training with other desiderata. For example, using accuracy and REV, trained models can select rationales that not only have predictive power but also include extra information which might explain the reasoning process.


**Overall Recommendation:**

Overall, the paper is well written and the motivation is clearly clarified, with reasonable metric proposed for free-text rationales.

**Review Confidence:**

3: The reviewer is fairly confident that the evaluation is correct

---

### Decision · Program_Chairs · 2022-10-23

**Decision:**

Accept

**Comment:**

Following the unanimous recommendations from reviewers, the submission is accepted.